# High-Dose Vitamin D Supplementation Significantly Affects the Placental Transcriptome

**DOI:** 10.3390/nu15245032

**Published:** 2023-12-07

**Authors:** Anna Louise Vestergaard, Matilde K. Andersen, Rasmus V. Olesen, Pinar Bor, Agnete Larsen

**Affiliations:** 1Department of Obstetrics and Gynecology, Randers Regional Hospital, 8930 Randers, Denmarkisipinbo@rm.dk (P.B.); 2Department of Clinical Medicine, Aarhus University, 8200 Aarhus, Denmark; 3Department of Biomedicine, Aarhus University, 8000 Aarhus, Denmarkal@biomed.au.dk (A.L.); 4Department of Obstetrics and Gynecology, Aarhus University Hospital, 8200 Aarhus, Denmark

**Keywords:** vitamin D, placenta, immune function, pregnancy, NGS

## Abstract

Vitamin D deficiency is a highly prevalent obstetrical concern associated with an increased risk of complications like pre-eclampsia, gestational diabetes, and growth retardation. Vitamin D status in pregnancy is also linked to long-term offspring health, e.g., the risk of obesity, metabolic disease, and neurodevelopmental problems. Despite the suspected role of vitamin D in placental diseases and fetal development, there is limited knowledge on the effect of vitamin D on placental function. Thus, we performed next-generation RNA sequencing, comparing the placental transcriptome from uncomplicated term pregnancies receiving the often-recommended dose of 10 µg vitamin D/day (*n* = 36) with pregnancies receiving 90 µg/day (*n* = 34) from late first trimester to delivery. Maternal vitamin D status in the first trimester was also considered. We found that signaling pathways related to cell adhesion, immune function, and neurodevelopment were affected, supporting that increased vitamin D supplementation benefits placental function in established pregnancies without severe vitamin D deficiency, also underlining the importance of vitamin D in brain development. Specific effects of the first trimester vitamin D status and offspring sex were also identified. Further studies are warranted, addressing the optimal vitamin status during pregnancy with a focus on organ-specific vitamin D needs in individual pregnancies.

## 1. Introduction

Despite numerous studies linking vitamin D deficiency to long-term health issues in offspring [1,2,3,4] and an increased risk of pregnancy complications like preeclampsia (PE), fetal growth retardation (FGR), and preterm birth [5,6,7,8,9,10,11,12,13,14], vitamin D deficiency remains common among pregnant women in both high- and low-income countries [15]. Placental development and function are essential for pregnancy outcome and closely related to conditions such as PE and FGR [16,17]. Some have even coined a Placental Origin of Health and Disease Hypothesis [18], underlining that placental factors have a strong impact on human development [18,19,20]. Calcitriol (1,25(OH)_2_D), the active form of vitamin D, binds to the vitamin D receptor (VDR), which is abundantly present in placental tissue (15,16). Moreover, placental tissue contains the hydroxylating enzyme CYP27B1, enabling the final step of calcitriol formation, the conversion of calcidiol (25(OH)D) into calcitriol (1,25(OH)_2_D), to take place within the placenta itself [21,22]. However, the current knowledge of the placental functions sensitive to vitamin D deficiency and the placental benefits of increasing vitamin D supplementation during pregnancy remains limited. Beneficial effects of increased maternal vitamin D supplementation on several pregnancy complications, including PE, have been found in a few small interventional studies [23,24,25], whereas Hossain et al. [26] found no effect of a 100 µg vitamin D supplementation intervention in 193 pregnant women. However, none of these studies focused on placental biology. More knowledge is thus needed to understand the role of vitamin D in the placental function and help identify risk pregnancies in need of higher doses of vitamin D supplementation. Vitamin D deficiency is defined as serum 25(OH)D <50 nmol/L [27], but in pregnancy, 25(OH)D concentrations ≥75 nmol/L are considered as vitamin D sufficiency [28], and some have even suggested that the 25(OH)D serum concentration should be in excess of 100 nmol/L [23]. Today, most countries recommend a vitamin D supplement for pregnant women ranging from 10 to 20 µg; however, this is often not enough to reach sufficiency in terms of a 25(OH)D concentration ≥75 nmol/L [29].

In this study, we present, as some of the first, next-generation RNA sequencing (NGS) data on term placental tissue comparing how the placental function differs following two different doses of maternal vitamin D supplementation in pregnancy. The aim of this study was to investigate alterations in the placental transcriptome at term following a high dose (90 µg) of vitamin D supplementation during pregnancy compared to the currently recommended dose (10 µg) of vitamin D supplementation. 

## 2. Materials and Methods

### 2.1. Study Setting and Study Population

The placental tissue was obtained at the time of birth from 70 singleton pregnancies from the ongoing clinical trial GRAVITD, described in detail elsewhere [30], and registered at ClinicalTrial.gov on 17 February 2020 (NCT04291313). The participants were enrolled when attending a nuchal translucency scan at Randers Regional Hospital. This scan is a part of the prenatal screening program in Denmark offered to all pregnant women free of charge. At the time of enrolment, the participants were in gestational week 11–16. Upon enrolment, a blood sample was collected along with a questionnaire describing lifestyle habits, including the use of medication. 

### 2.2. Intervention

Participants were randomized in a 1:1 allocation ratio into two groups receiving: (1) 10 µg vitamin D3 daily (control, current recommendation during pregnancy in Denmark) or (2) 90 µg of vitamin D3 daily (intervention). Both the participants and the laboratory personnel were blinded to the assigned dose, and the participants were instructed not to take any other vitamin D supplements. A prenatal multivitamin supplementation was given alongside vitamin D, and the participants were encouraged to take a 40–50 mg iron supplement as well. During the 3rd trimester, one half of the participants were invited to provide a new blood sample and complete a new questionnaire. The blood samples were analyzed for 25(OH)D using high-performance liquid chromatography coupled with tandem mass spectrometry at the Department of Biochemistry at Aarhus University Hospital. 

### 2.3. Collection of Placental Tissue

At the time of delivery, the placenta was placed at 5 °C by a midwife, and systematically selected random samples of villus tissue were collected by a member of the research team within 5 h after delivery. The tissue samples used in this study were collected from February 2021 to January 2022. Exclusion criteria were (1) placental tissue from preterm birth (<37 weeks) or multiple pregnancies; (2) placental tissue from pregnancies in which preeclampsia (PE), gestational diabetes (GDM), or intrauterine growth restriction (IUGR) occurred; and (3) placental tissue from pregnancies during which the mother was smoking or using antidepressants, medicine for thyroid disorders (levothyroxine), sumatriptan, medicine affecting the immune system (azathioprine, acyclovir, mesalazine), or receiving hormone treatment like progesterone or estradiol. The sample selection was randomized by listing all samples according to their record id in each group (10 µg vs. 90 µg) and selecting every other sample collected until a total of 70 samples were reached. If a pregnancy was excluded due to the exclusion criteria, the next eligible sample on the list was selected as a replacement. 

### 2.4. Random Sample Collection

Each placenta was divided into 12 parts. The place of sampling was randomly determined using a 12-sided dice, and a villus sample was obtained from the central part of the selected slice after the removal of the decidua. The tissue was rinsed in isotonic saline and placed in RNA, later followed by storage at −20 °C until further analysis.

### 2.5. RNA Purification

Total RNA was extracted from 30 mg placental tissue by Trizol (Ambion, Life Technologies, Roskilde, Denmark) and purified using the RNeasy Mini kit (Qiagen, Germantown, MD, USA). The RNA concentration and purity were determined by an absorbance measurement using the NanoDrop 2000 instrument (Thermo Scientific, Roskilde, Denmark), and the quality of the extracted RNA was evaluated by agarose gel electrophoresis.

### 2.6. Next-Generation RNA Sequencing

The library preparation and next-generation RNA sequencing were performed by the NGS Core Center, Department of Molecular Medicine, at Aarhus University Hospital, Denmark, according to the manufacturer’s protocol (Illumina, San Diego, CA, USA). A total RNAseq library preparation was conducted using KAPA RNA HyperPrep Kits with RiboErase. A total RNA sequencing assembling ~50 M 150 paired-end read pairs per sample was conducted on the Illumina NovaSeq 6000 (Illumina, San Diego, CA, USA). All samples were processed in the same manner, with all sequencing libraries created in the same batch and sequenced together.

### 2.7. Data Preparation and Statistics

Reads were aligned to the human reference genome (GRCh38.p14) using Hisat2 version 2.2.1, and the quantification of counts was performed using FeatureCounts version 2.0.3. The differentially expressed genes (DEGs) were determined from count tables using DESeq2 version 1.38.0 and reported after a Benjamini–Hochberg false discovery rate (FDR) (10%) correction. Nominally significant DEGs with a *p*-value < 0.001 were used as the input for the gene set enrichment analyses using FUMA [31] and ShinyGO version 0.77, as well as for transcription factor binding site enrichment analysis using CiiiDER [32]. Furthermore, to utilize the full dataset, a gene ontology enrichment analysis was also conducted for nominally significant DEGs with a *p*-value < 0.05. Data preparation and the statistical analysis of the RNA sequencing data were performed by the Bioinformatic Core Facility at the Department of Biomedicine, Aarhus University in collaboration with the research group.

Demographic data and the clinical characteristics from delivery were described as mean ± SD for continuous variables and as numbers and frequencies for categorical variables. For mean 25(OH)D concentrations, the 95%CI was also calculated. Student’s *t*-test for continuous variables and a chi^2^-test for categorical variables were used to evaluate if there were any differences between the women in the two vitamin D dosing groups (90 µg vs. 10 µg). A statistical significance level of <0.05 was used if not otherwise stated. The descriptive statistics were performed using STATA (version 18, StataCorp, College Station, TX, USA).

### 2.8. Ethical Approval

The GRAVITD trial and this ancillary study of NGS of placental tissues were approved by the scientific ethical committee of the Central Denmark Region on 9 September 2019 (1–10-72–54-19). Prior to enrolment, all participants received both written and oral information about the clinical trial and the study of placental function before giving their written informed consent, in accordance with the principles of the Declaration of Helsinki.

## 3. Results

### 3.1. Sample Characteristics

A total of 70 placental samples obtained from singleton term pregnancies were used for this study. In total, 36 of the women had been randomized to receive 10 µg of vitamin D during the last six months of pregnancy, whereas 34 women received a total of 90 µg vitamin D during this period. The main demographic data on the pregnancies in each group are described in Table 1. The groups did not differ in any demographic data category at baseline or in any of the clinical characteristics related to birth, including age, BMI, mode of delivery, gestational age at delivery, or offspring sex. Upon enrolment, the average vitamin D concentration (25(OH)D) was 69.9 (95%CI: 61.9; 78.0) nmol/L (90 µg group) vs. 76.4 (95%CI: 70.1; 82.7) nmol/L (10 µg group), *p* = 0.203. Vitamin D (25(OH)D) was measured during the third trimester in a total of 42 women. As expected, this analysis revealed a statistically significant higher vitamin D concentration in the women who received the highest dose of supplementation, i.e., 124.2 (95%CI: 112.9; 135.5) nmol/L (90 µg, *n* = 23) vs. 86.5 (95%CI: 76.5; 96.5) nmol/L (10 µg, *n* = 19), *p* < 0.0001. 

### 3.2. Gene Expression Profiling

We characterized the overall differences between the gene expression profiles of the 70 samples, utilizing an unsupervised classification method—Principal Component Analysis (PCA). The PCA using normalized gene expressions revealed two distinct clusters, corresponding to the sex of the offspring resulting from the pregnancies along the first principal component 1 (PC1) (Figure 1). Notably, the PC1 explained up to 17% of the variation, while the PC2 explained 15% of the variation, demonstrating that the sex of the offspring significantly impacts the placental gene expression. None of the other sample metadata contributed notably to the first five principal components. Consequently, we controlled for sex in our subsequent analyses.

First, we set out to assess if the maternal serum 25(OH)D concentration measured during placental development was associated with measurable molecular changes in the delivered placenta. Particularly, we divided samples according to their first trimester vitamin D status measured upon enrolment into the clinical trial. We compared the women with a sufficient 25(OH)D concentration ≥75 nmol/L to the women who were vitamin D-deficient/insufficient in terms of a 25(OH)D concentration <75 nmol/L and conducted a differential expression (DE) test between the two groups. Showing a solid impact of the first trimester maternal serum 25(OH)D concentration on the placental transcriptome, a broad range of nominally significant DEGs were found (*p* < 0.001). Reflecting the heterogeneity of the samples evident from the PCA, only six genes were, however, identified as differentially expressed following false discovery correction at 10% (Figure 2 and Appendix A). Interestingly, these six genes included *PTH*, encoding the parathyroid hormone—a key regulator of the vitamin D metabolism [33]—and *C2* encoding the complement component 2, a regulator of the complement system which has previously been connected to vitamin D status [34]. Further supporting the existence of a functional connection, an enrichment analysis of the transcription factor binding site in the promotor sequence of nominally significant DEGs compared to non-DEGs showed the most significant enrichment for the binding site of the Vitamin D-binding protein (VDBP) (*p* = 0.004).

Next, we proceeded to examine if providing pregnant women with increased vitamin D supplementation (90 µg/day) resulted in a measurable change in placental gene expression compared to the currently recommended dose of 10 µg/day. Three genes emerged as differentially expressed following false discovery correction at 10% (Figure 3a and Appendix A). This included *JPH1*, that has previously been reported to be differentially regulated following vitamin D exposure in the colon [35]. While only a few genes were significantly differentially expressed at a conservative adjusted *p*-value cut-off at 10%, hierarchical clustering based on the expression of nominally significant DEGs with *p* < 0.001 convincingly clustered samples according to the assigned vitamin D dosing group (Figure 3b). Hence, this prompted us to perform a gene set enrichment analysis on this gene set to provide a functional interpretation of the placental transcriptomic impact of vitamin D supplementation. This analysis revealed that only one KEGG pathway was affected, i.e., cell adhesion molecules. Thus, we also performed a gene set enrichment analysis for all nominally significant DEG with a *p* < 0.05 to obtain a broader insight into functional changes following increased vitamin D supplementation. All the identified pathways in this gene set were related to the immune function (Figure 3c). Highlighting the putative clinical relevance of the identified transcriptomic changes, the DEG list was significantly enriched with genes containing genetic variants associated with traits like type 1 diabetes, autism spectrum disorder, schizophrenia, and age-related macular degeneration (Figure 3d). 

As we show that the sex of the offspring significantly affects placental gene expression, we speculated whether vitamin D supplementation during pregnancy had different effects on placental function based on the sex of the fetus. By splitting samples based on the sex of the offspring, we found four DEGs in the placental tissue from pregnancies with male offspring and one DEG in the placental tissue from pregnancies with female offspring in the 90 µg vitamin D group, while ~50 nominally significant DEGs at *p* < 0.001 were identified in each group (Figure 4).

## 4. Discussion

The impact of vitamin D on placental function has not previously been explored, even though impaired placental function is involved in vitamin D deficiency-associated conditions like PE and FGR [5,7,8,9,10,29]. Here, we aim to address this knowledge gap by conducting a state-of-the-art NGS study involving placental villus tissue from a randomized clinical trial. We demonstrate distinct differences in placental gene expression comparing the response of a 90 µg maternal vitamin D supplementation to a dose of 10 µg, which is the commonly recommended dose to pregnant women today. Overall, signaling pathways related to cell adhesion molecules and pathways related to the immune system, mainly T-cell function, were affected by the increased vitamin D load from gestational week 11–16 and onwards. Additional placental differences were detectable when examining the placental transcriptome based on the initial maternal vitamin D status. This confirms that vitamin D is not only essential for placental growth and function, but maternal vitamin D status in early pregnancy was also linked to the *XIST* gene regulating X chromosome inactivation [36] as well as to the specific immune function regulated by the *C2*, *SKIC2*, and *SQSTM1* genes [37]. 

Among the statistically significant findings was an upregulation of *EIF3K.* The encoding protein EIF3K plays an important role in protein synthesis [38] and is believed to suppress apoptosis and inflammatory processes [39], thus supporting that vitamin D upholds placental function in a broader sense. EIF3K is known to plays a regulatory role in modulating the expression of CD138 (syndecan-1) [40]. CD138-positive cells include cell types associated with immune function, such as the CD138-positive fraction of plasma cells [40,41]. Moreover, syndecan 1 is abundant in the syncytiotrophoblast and known to play a role in cell adhesion [41]. This link between vitamin D and cell adhesion could have clinical implications as cell adhesion molecules play a pivotal role in the process of placentation [42]. Discrepancies in the expression of these molecules may potentially contribute to adverse outcome, such as miscarriage or PE [42].

A novel finding of this study was the link between vitamin D and *TDRD12*, a gene not previously associated with placental function, although it is known that *TDRD12* is important for germ cell function and fertility [43,44], and a lower expression of *TDRD12* is associated with azoospermia in human males [43]. *TDRD12* is also believed to play an important role in piRNA pathways and the regulation of transposable elements [44,45,46], which are believed to be highly important for placental function and could potentially provide another link to the development of PE [47].

Our gene set enrichment analysis showed an effect on the signaling pathways involved in immune response. A regulatory effect of vitamin D on the immune system in general is well described [48,49,50,51]. Disruption of immunogenic tolerance is believed to be a part of the pathogenesis in PE patients, leading to abnormal placentation [52], and PE patients have increased concentrations of proinflammatory cytokines, e.g., TNF-alfa and IL-6, and decreased concentrations of anti-inflammatory cytokines like IL-10 [53]. Moreover, the microarray analysis of maternal blood cells obtained in early pregnancy showed that vitamin D-associated genes relate to both PE [54] and spontaneous preterm birth [55]. Most of the vitamin D-associated genes involved in PE were associated with maternal systemic changes in immune system processes affecting both the innate and adaptive immune response [54]. Overall, this indicates that the maternal inflammatory status is important for pregnancy outcome. It is worth noting that the present study is the first study to show a direct impact of high vitamin D supplementation during pregnancy on placental immune function. Notably, our findings suggest that vitamin D status in early pregnancy independently influences the placental transcriptome. It is possible that we would have seen a more distinct effect on the placental transcriptome if the increased vitamin D supplementation had been initiated earlier in pregnancy or maybe even before conception. 

When interpreting the data, it is important to consider that we did not include placental samples from women diagnosed with conditions such as PE, FGR, or GDM. Therefore, further studies are needed to identify vitamin D-sensitive pathways differentially expressed in diseased placental tissue which could be targeted to limit disease risk. Another limitation to our study is the relative low prevalence of vitamin D deficiency (25(OH)D < 50 nmol/L) at baseline. It is possible that other or more pronounced effects of increased maternal vitamin D supplementation could have been seen on the placental transcriptome in a population with a poorer vitamin D status at baseline. A strength of this study is the randomized design securing two different groups of placental tissue from pregnancies included evenly throughout the year, thus minimizing the risk of confounders. Notably, our data were analyzed as intention to treat, and we cannot be certain that all participants used the allocated supplement daily as we did not perform pill counts. 

Vitamin D deficiency in uncomplicated pregnancies is associated with long-term health difficulties, including asthma, type 1 diabetes, multiple sclerosis, and respiratory infections in the affected children [2,4,56,57], which are all conditions related to the immune function of the offspring. Whether these effects are more likely to be caused by placental changes or occur as a consequence of an insufficient vitamin D supply to the fetus remains to be clarified. However, we did find genes related to type 1 diabetes and other autoimmune diseases in a gene ontology enrichment analysis, which supports a potential association between vitamin D, placenta, and autoimmune diseases later in life for the offspring. 

Neurodevelopmental disorders like autism spectrum disorders, attention deficit hyperactivity disorder, and schizophrenia have previously been associated with intrauterine vitamin D deficiency [2,3]. Supporting these findings, our gene set enrichment analysis further identified six genes involved in pathways linked to autism spectrum disorders and schizophrenia. Interestingly, animal studies have directly linked placental inflammation to neurodevelopment as inflammatory signals reach the developing brain [58,59]. Furthermore, placental inflammation is associated with a reduced number of dendritic processes in the fetal brain, resulting in impaired learning [59]. Supportive of the role of vitamin D in neurodevelopment, a very recent study by Rodgers et al. [60] found that higher maternal vitamin D supplementation during pregnancy and a higher maternal 25(OH)D concentration led to better language development in the offspring after controlling for multiple factors, including maternal education.

Conducting the analyses of the RNA sequencing data, we saw a substantial impact of fetal sex, indicating a potential sex-related difference in the placental vitamin D response. A sex-dependent transcriptome difference has previously been observed in response to vitamin D supplementation in a study on pre-diabetic adults by Pasing et al. [61]. Moreover, pregnancy complication associated with vitamin D deficiency, such as preterm birth, hypertensive disorders of pregnancy, and GDM, are more common in pregnancies with male fetuses [62]. However, further studies are warranted to determine whether sex-dependent differences in placental function should be taken into consideration when addressing the effect of maternal vitamin D supplementation.

## 5. Conclusions

In conclusion, our study demonstrates that increased maternal vitamin D supplementation during the second and third trimester of pregnancy leads to alterations in gene expression and in signaling pathways related to the immune system as well as pathways leading to autoimmune diseases, autism spectrum disorders, and schizophrenia. These findings support the premise that even healthy, uncomplicated pregnancies could benefit from increased vitamin D supplementation during pregnancy. Moreover, the vitamin D status in early pregnancy has an effect of placental gene expression at term, and early initiation of vitamin D supplementation may hold additional benefits through the effect on cell adhesion molecules crucial for placentation. Further research is needed to determine the optimal level of 25(OH)D throughout pregnancy in order to support both mother, child, and placental function.

## Figures and Tables

**Figure 1 nutrients-15-05032-f001:**
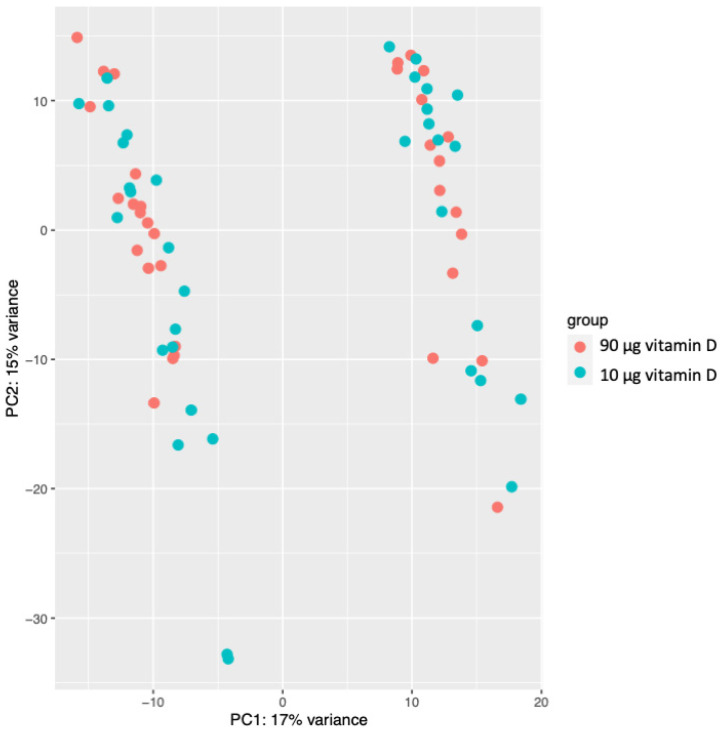
Principal component analysis of normalized expression in all 70 samples.

**Figure 2 nutrients-15-05032-f002:**
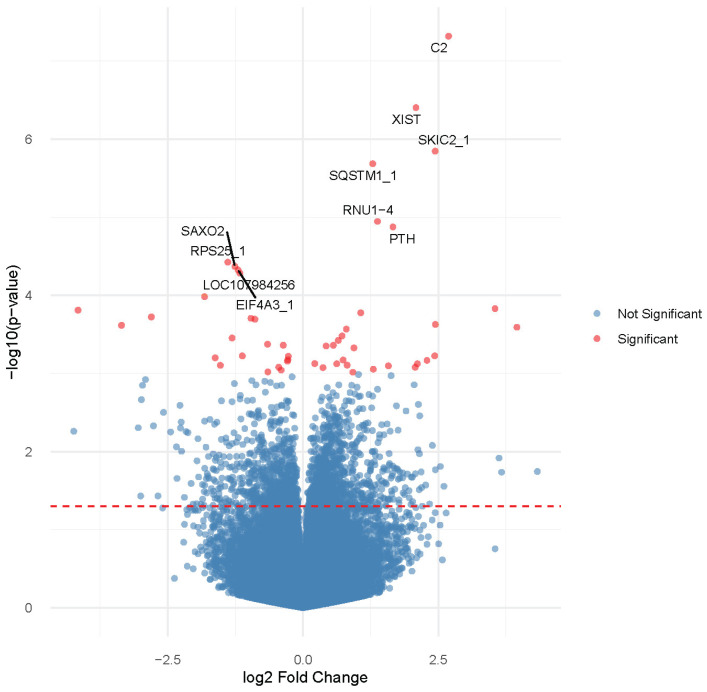
Volcano plot of differential expression analysis between the vitamin D-deficient/insufficient group (*n* = 28) against the vitamin D-sufficient group (*n* = 40) with correction for offspring sex. Highlighted in red are nominally significant DEGs with *p* < 0.001. Top ten most significant DEGs are marked with gene symbol labels, incl. six DEGs that remain significant after false discovery correction at 10% (*C2*, *XIST*, *SKIC2*, *SQSTM1*, *RNU1-4*, and *PTH*). The red dashed line represents a nominally significant cut-off of *p* = 0.05.

**Figure 3 nutrients-15-05032-f003:**
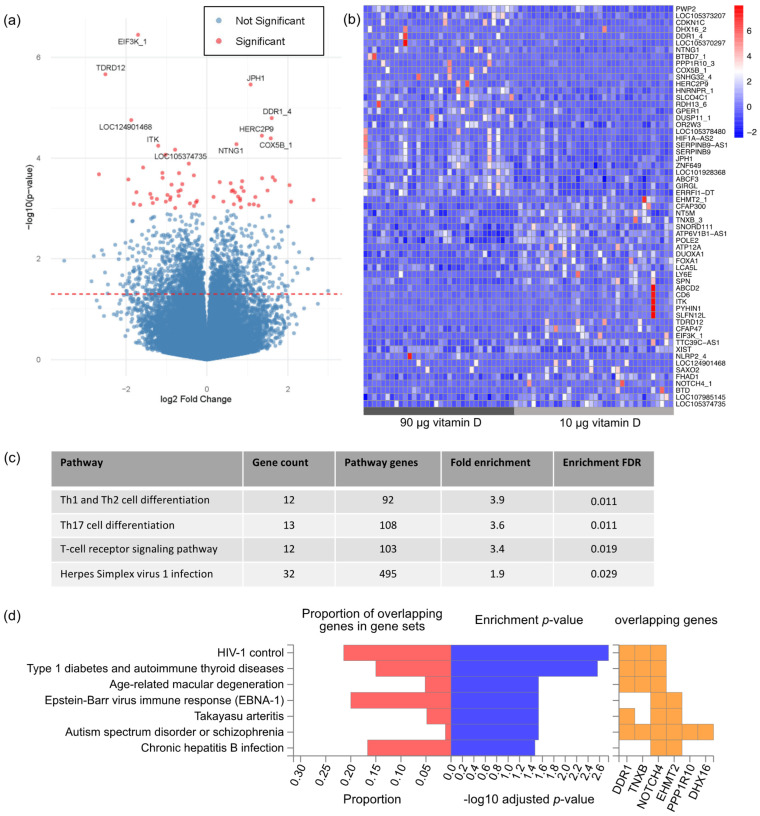
Differential expression analysis between the 90 µg vitamin D group and the 10 µg group with correction for sex. (**a**) Volcano plot of differential expression analysis. Highlighted in red are nominally significant DEGs with *p* < 0.001. Top ten most significant DEGs are marked with gene symbol labels, incl. three DEGs that remain significant after false discovery correction at 10% (*EIF3K*, *TDRD12*, and *JPH1*). The red dashed line represents a nominally significant cut-off of *p* = 0.05. (**b**) Heatmap of normalized expression values of nominally significant DEGs with *p* < 0.001 with hierarchical clustering applied. (**c**) Gene set enrichment analysis of nominally significant DEGs with *p* < 0.001 against GO terms. (**d**) Gene set enrichment analysis against the GWAS catalogue.

**Figure 4 nutrients-15-05032-f004:**
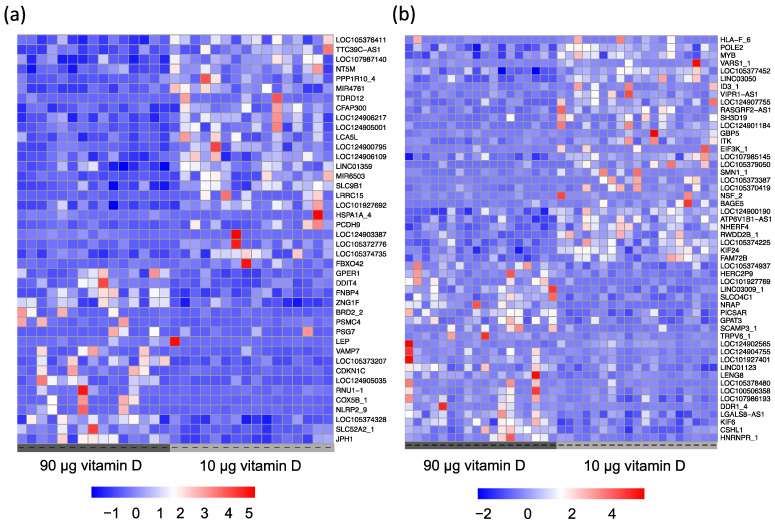
Heatmaps of normalized expression values of the nominally significant DEGs with *p* < 0.001 with hierarchical clustering applied for (**a**) placental samples from pregnancies with male offspring (42 DEGs) and (**b**) placental samples from pregnancies with female offspring (52 DEGs).

**Table 1 nutrients-15-05032-t001:** Demographic data and clinical characteristics of pregnant women in the two vitamin D dosing groups.

	90 µg of Vitamin D *n* = 34	10 µg of Vitamin D *n* = 36	*p*-Value
**Maternal age (years), *mean* (*± SD)***	30.4 (5.2)	29.1 (4.1)	0.237
**BMI (kg/m^2^) *mean* (±*SD*)**	24.8 (4.8)	25.9 (5.9)	0.372
**Parity *n* (%)**			0.520
**Nulliparous**	17 (50.0)	16 (44.4)	
**Primiparous**	10 (29.4)	15 (41.7)	
**Multiparous**	7 (20.6)	5 (13.9)	
**Season at enrolment *n* (%)**			0.814
**Winter**	9 (26.5)	7 (19.4)	
**Spring**	9 (26.5)	8 (22.2)	
**Summer**	8 (23.5)	11 (30.6)	
**Autumn**	8 (23.5)	10 (27.8)	
**Gestational age at enrolment (week + day) *mean* (±*SD*)**	12 + 6 (0.8)	12 + 6 (1.1)	0.938
**First trimester 25(OH)D (nmol/L) *mean* (±*SD*)**	69.9 (22.8)	76.4 (18.3)	0.203
**Third trimester 25(OH)D (nmol/L) *mean* (±*SD*) ***	124.2 (26.2)	86.5 (20.7)	<0.0001
**First trimester vitamin D status *n* (%)**			0.074
Deficient < 50 nmol/L	10 (30.3)	3 (8.6)	
Insufficient 50–74 nmol/L	6 (18.2)	9 (25.7)	
Sufficient ≥ 75 nmol/L	17 (51.5)	23 (65.7)	
**Third trimester vitamin D status *n* (%) ***			0.059
Deficient < 50 nmol/L	0 (0.0)	1 (5.3)	
Insufficient 50–74 nmol/L	1 (4.3)	5 (26.3)	
Sufficient ≥ 75 nmol/L	22 (95.7)	13 (68.4)	
**Gestational age at delivery (week + day), *mean* (±*SD*)**	39 + 6 (1.0)	40 + 0 (1.2)	0.395
**Mode of delivery *n* (%)**			0.313
**Vaginal**	28 (82.4)	26 (72.2)	
**Caesarean section**	6 (17.7)	10 (27.8)	
**Sex of the baby *n* (%)**			0.794
**Female**	19 (55.9)	19 (52.8)	
**Male**	15 (44.1)	17 (47.2)	

* A third trimester serum sample was available from 42 participants, 23 from the 90 µg vitamin D group, and 19 from the 10 µg vitamin D group as only half of the women in the GRAVITD trial had a second serum sample collected during the study.

## Data Availability

The full dataset is not publicly available as we do not have consent from the participants to publish the full dataset. However, a de-identified dataset will be available from the corresponding author upon reasonable request.

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
