# Peer review of "High-Dose Vitamin D Supplementation Significantly Affects the Placental Transcriptome"

_nutrients, 2023, doi:10.3390/nu15245032_

Round 1
Reviewer 1 Report
Comments and Suggestions for Authors
The effects of vitamin D deficiency on placental transcription was studied in this manuscript. This paper is well written and comprehensive on the effects of vitamin D deficiency in pregnancy. There are a bit of issues should be solved.
1. Format errors in some statements, such as missing punctuation in line 73, 75, 282 , redundant spaces in line 285,319 and case incorrect in line 228.
2. There are errors in the format of references, for example, the year needs to be bold.
3. In figure 1, the image of molecules is unclear and the font is blurred, it is necessary to improve the pixel of the image.
Author Response
The effects of vitamin D deficiency on placental transcription was studied in this manuscript. This paper is well written and comprehensive on the effects of vitamin D deficiency in pregnancy. There are a bit of issues should be solved.
- Format errors in some statements, such as missing punctuation in line 73, 75, 282 , redundant spaces in line 285,319 and case incorrect in line 228.
We thank the reviewer for the thorough revision. These errors have now been corrected.
- There are errors in the format of references, for example, the year needs to be bold.
The errors in the format of references have been corrected.
- In figure 1, the image of molecules is unclear and the font is blurred, it is necessary to improve the pixel of the image.¨
A new and improved figure 1 has been added as requested.
Reviewer 2 Report
Comments and Suggestions for Authors
A minor suggestion is to correct the use of gender when referring to sex ie male/female
Author Response
A minor suggestion is to correct the use of gender when referring to sex ie male/female
We agree with the reviewer that this is not the proper phrasing and the figure legend for figure 3 have been corrected, so only the term sex is used throughout the text when referring to male/female.
Reviewer 3 Report
Comments and Suggestions for Authors
Vestergaard et al. explore the impact of high-dose vitamin D supplementation on the functioning of the placenta during pregnancy. Past research gives some evidence linking a deficiency in vitamin D to a range of obstetrical challenges and long-term health issues in offspring, including conditions like pre-eclampsia, gestational diabetes, growth retardation, obesity, metabolic diseases, and neurodevelopmental disorders. This study employed RNA sequencing to analyze and compare placental transcriptomes in two distinct groups of term pregnancies. One group received the standard recommended vitamin D dosage of 10µg per day, while the other was administered a higher dosage of 90µg per day, starting from the late first trimester until delivery. Findings from the study highlight that enhanced vitamin D supplementation positively influences placental function, impacting key pathways involved in cell adhesion, immune response, and neurodevelopment. These results underscore the critical role of vitamin D in both placental health and fetal development, indicating a need for further research into the specific vitamin D requirements during pregnancy, tailored to the organ-specific needs of individual pregnancies. This review paper is interesting. I am quite open to looking at a revised version if the authors could address some major and minor issues in a satisfactory fashion, which I describe in more detail below.
Major issues:
1. While the study demonstrates the effects of vitamin D on placental gene expression, it does not directly link these changes to specific clinical outcomes in mothers or offspring. Thus, the practical implications of these findings remain somewhat speculative. In other words, The study focuses on the effects of vitamin D supplementation during pregnancy, but it does not address the long-term effects of altered gene expression on the health of the mother or child post-delivery. Do the authors have the disease outcomes to explore how the difference in the gene expression explains the disease outcomes? If they don’t have the clinical outcomes, I would like to suggest they add a discussion paragraph to acknowledge this limitation.
2. The resolution of all figures is pretty low. They all look blurred to me. Please ensure that the main figures are exported in a higher resolution. In addition, some figures have really small font sizes. For instance, the axis labels in Figure 1 (i.e., PC1 and PC2) are too small. Please increase the font sizes when necessary.
3. On lines 207-208, I think the citation is not properly showed because it displays “Error!Reference source not found”.
4. I saw many grammatical errors, with some of them listed below in the minor comments. Please correct them and other ones I did not detect.
Minor comments:
1. Lines 17&251: “recommend dose” -> “recommended dose”
2. Lines 19&23: “first trimester” -> “the first trimester”
3. Lines 24-25: “with focus on organ specific” -> “with a focus on organ-specific”
4. Line 53: “reach sufficiency” -> “reach a sufficiency”
5. Line 58: “high dose (90 μg) vitamin D” -> “a high dose (90 μg) of vitamin D”
6. Line 63: “Placental tissue were obtained at time of birth” -> “Placental tissue was obtained at the time of birth”
7. Line 135: “continues variables” -> “continuous variables”
8. Line 154: “last six month of pregnancy” -> “last six months of pregnancy”
9. Line 158: “average vitamin D concentration” -> “the average vitamin D concentration”
10. Line 167: “A third trimester serum sample were” -> “A third-trimester serum sample was”
11. Line 175: “PC1 explained up towards 17%” -> “PC1 explained up to 17%”
12. Line 176: “sex significantly impact” -> “sex significantly impacts”
13. Line 210: “While only few genes” -> “While only a few genes”
14. Line 292: “influences on the placental transcriptome” -> “influences the placental transcriptome”
15. Line 328: “A sex dependent transcriptome” -> “A sex-dependent transcriptome”
Comments on the Quality of English LanguageI saw many grammatical errors, with some of them listed in my minor comments. An extensive English edits are required.
Author Response
- While the study demonstrates the effects of vitamin D on placental gene expression, it does not directly link these changes to specific clinical outcomes in mothers or offspring. Thus, the practical implications of these findings remain somewhat speculative. In other words, The study focuses on the effects of vitamin D supplementation during pregnancy, but it does not address the long-term effects of altered gene expression on the health of the mother or child post-delivery. Do the authors have the disease outcomes to explore how the difference in the gene expression explains the disease outcomes?If they don’t have the clinical outcomes, I would like to suggest they add a discussion paragraph to acknowledge this limitation.
The reviewer rightly noticed that placental samples from complicated pregnancies could give additional information beyond the scope of this study, and that such studies would indeed be valuable. This manuscript is based on analysis of placental sample from uncomplicated pregnancies a fact we have now further emphasized in the text. Accordingly, we cannot say anything about the direct impact on pregnancy complications or indeed later life diseases l, thus we must limit ourselves to debate our findings in the context of current knowledge and associations with clinical outcome is of cause speculative. These limitations have been further clarified in the discussion to avoid misunderstandings.
However, as soon as all participants in the GRAVITD trial have delivered, we will present the clinical effects on complications, and we hope to have a sufficient number of placentas from complicated pregnancies at least in terms of PE and GDM to allow us to do further NGS studies and compare to what we found in uncomplicated pregnancies.
- The resolution of all figures is pretty low. They all look blurred to me. Please ensure that the main figures are exported in a higher resolution. In addition, some figures have really small font sizes. For instance, the axis labels in Figure 1 (i.e., PC1 and PC2) are too small. Please increase the font sizes when necessary.
As suggested, we have now added updated versions of the figures with higher resolution and enhanced font size to the manuscript.
- On lines 207-208, I think the citation is not properly showed because it displays “Error!Reference source not found”.
We apologize – this was not evident in our version of the submitted manuscript. This formatting error has now been corrected.
- I saw many grammatical errors, with some of them listed below in the minor comments. Please correct them and other ones I did not detect.
We thank the reviewer for the in-depth revision and have corrected accordingly, also by having the manuscript evaluated by someone fluent in written English in order to correct grammatical errors.
Minor comments:
- Lines 17&251: “recommend dose” -> “recommended dose”
- Lines 19&23: “first trimester” -> “the first trimester”
- Lines 24-25: “with focus on organ specific” -> “with a focus on organ-specific”
- Line 53: “reach sufficiency” -> “reach a sufficiency”
- Line 58: “high dose (90 μg) vitamin D” -> “a high dose (90 μg) of vitamin D”
- Line 63: “Placental tissue were obtained at time of birth” -> “Placental tissue was obtained at the time of birth”
- Line 135: “continues variables” -> “continuous variables”
- Line 154: “last six month of pregnancy” -> “last six months of pregnancy”
- Line 158: “average vitamin D concentration” -> “the average vitamin D concentration”
- Line 167: “A third trimester serum sample were” -> “A third-trimester serum sample was”
- Line 175: “PC1 explained up towards 17%” -> “PC1 explained up to 17%”
- Line 176: “sex significantly impact” -> “sex significantly impacts”
- Line 210: “While only few genes” -> “While only a few genes”
- Line 292: “influences on the placental transcriptome” -> “influences the placental transcriptome”
- Line 328: “A sex dependent transcriptome” -> “A sex-dependent transcriptome”
Comments on the Quality of English Language
I saw many grammatical errors, with some of them listed in my minor comments. An extensive English edits are required.
The manuscript has now been corrected by someone fluent in written English in order to avoid grammatical errors, including the ones carefully pointed out by the reviewer.
Round 2
Reviewer 2 Report
Comments and Suggestions for Authors
Thank you for your resubmission. I have no further comments. Well done.
Reviewer 3 Report
Comments and Suggestions for Authors
The authors have answered all the questions that I raised last time. I do not have further comments.